# Physicochemical Characterization of Bilayer Hybrid Nanocellulose-Collagen as a Potential Wound Dressing

**DOI:** 10.3390/ma13194352

**Published:** 2020-09-30

**Authors:** Kai Shen Ooi, Shafieq Haszman, Yon Nie Wong, Emillia Soidin, Nadhirah Hesham, Muhammad Amirul Arif Mior, Yasuhiko Tabata, Ishak Ahmad, Mh Busra Fauzi, Mohd Heikal Mohd Yunus

**Affiliations:** 1Department of Physiology, Faculty of Medicine, Universiti Kebangsaan Malaysia Medical Centre, Jalan Yaacob Latif, Bandar Tun Razak, Kuala Lumpur 56000, Malaysia; a156122@siswa.ukm.edu.my (K.S.O.); a152989@siswa.ukm.edu.my (S.H.); a156216@siswa.ukm.edu.my (Y.N.W.); a153090@siswa.ukm.edu.my (E.S.); ga02099@siswa.ukm.edu.my (N.H.); p94583@siswa.ukm.edu.my (M.A.A.M.); 2Centre for Tissue Engineering and Regenerative Medicine, Faculty of Medicine, Universiti Kebangsaan Malaysia Medical Centre, Jalan Yaacob Latif, Bandar Tun Razak, Kuala Lumpur 56000, Malaysia; fauzibusra@ukm.edu.my; 3Department of Biomaterials, Institute for Frontier Medical Sciences, Kyoto University, 53 Kawara-cho Shogoin, Sakyo-ku Kyoto 606-8507, Japan; yasuhiko@infront.kyoto-u.ac.jp; 4School of Chemical Sciences and Food Technology, Faculty of Science & Technology, Universiti Kebangsaan Malaysia, Selangor 43600, Malaysia; gading@ukm.edu.my

**Keywords:** collagen, nanocellulose, hybrid bioscaffold, wound dressing

## Abstract

The eminent aim for advance wound management is to provide a great impact on the quality of life. Therefore, an excellent strategy for an ideal wound dressing is being developed that eliminates certain drawbacks while promoting tissue regeneration for the prevention of bacterial invasion. The aim of this study is to develop a bilayer hybrid biomatrix of natural origin for wound dressing. The bilayer hybrid bioscaffold was fabricated by the combination of ovine tendon collagen type I and palm tree-based nanocellulose. The fabricated biomatrix was then post-cross-linked with 0.1% (*w/v*) genipin (GNP). The physical characteristics were evaluated based on the microstructure, pore size, porosity, and water uptake capacity followed by degradation behaviour and mechanical strength. Chemical analysis was performed using energy-dispersive X-ray spectroscopy (EDX), Fourier transform infrared spectrophotometry (FTIR), and X-ray diffraction (XRD). The results demonstrated a uniform interconnected porous structure with optimal pore size ranging between 90 and 140 μm, acceptable porosity (>70%), and highwater uptake capacity (>1500%). The biodegradation rate of the fabricated biomatrix was extended to 22 days. Further analysis with EDX identified the main elements of the bioscaffold, which contains carbon (C) 50.28%, nitrogen (N) 18.78%, and oxygen (O) 30.94% based on the atomic percentage. FTIR reported the functional groups of collagen type I (amide A: 3302 cm^−1^, amide B: 2926 cm^−1^, amide I: 1631 cm^−1^, amide II: 1547 cm^−1^, and amide III: 1237 cm^−1^) and nanocellulose (pyranose ring), thus confirming the presence of collagen and nanocellulose in the bilayer hybrid scaffold. The XRD demonstrated a smooth wavy wavelength that is consistent with the amorphous material and less crystallinity. The combination of nanocellulose with collagen demonstrated a positive effect with an increase of Young’s modulus. In conclusion, the fabricated bilayer hybrid bioscaffold demonstrated optimum physicochemical and mechanical properties that are suitable for skin wound dressing.

## 1. Introduction

Wound dressing is used to stimulate and accelerate skin tissue repair. An ideal dressing should adhere well to the wound interface, maintain a balanced moist environment, allow gaseous exchanges, remove excess exudates, and act as a protective barrier against microorganisms [1,2]. To cater for all aspects of wound care treatment and its complexity, various models of a wound dressing with different morphologies and properties have been created such as alginate, collagen, and chitosan to be used as films, foams, hydrogels, hydrocolloids, or scaffolds. Nonetheless, for a chronic wound that deviates from the normal healing physiological stages, a further challenge is imposed on tailoring the wound dressing to meet the severity of the wound [2,3]. Recent studies have reported that chronic wounds pose an adverse impact worldwide whereby the prevalence rate in the developed countries is 1–2% of the general population, costing around 1–3% of the total healthcare expenditure [4,5]. The burden and demand for chronic wound care are expected to escalate substantially due to the growth of geriatric society and rising incidence of comorbidities [5]. Hence, a multi-competent dressing with better synergistic and complementary effects is to be expected for addressing the interdependent aspects within all stages of wound care management. These hybrid dressings are intended to treat superficial, partial, and full-thickness wounds, as well as acute or chronic wounds.

Being the most abundant protein in the human body and active constituent of extracellular matrix protein in the skin, collagen is a key component in wound healing [6]. In particular, collagen type I constitutes a pivotal key for cell–material interaction due to the presence of integrin-binding sites [7]. Thus, collagen was broadly used in tissue engineering due to its excellent biological properties, such as low immunogenicity, porous structure, good permeability, and biodegradability. In biomedical applications, collagen is commonly extracted from the tissue of animals with the most popular animals as the primary source being bovine and porcine [8]. However, the acceptability of collagen extracted from these animals remains a controversial subject in the Muslim and Hindu population since they are prohibited from consuming products of porcine and bovine origin. Despite the contravention of religious and cultural beliefs, Eriksson et al. (2013) concluded that the use of such extracted collagen is allowed provided that there are no other treatment modalities in a dire, critical life-threatening situation [9]. However, a conflict of interest occurs between the practitioner and patient which causes an ethical dilemma and religious distress to both parties. Therefore, in a multicultural and multiracial society, it is of the utmost importance to consider the religious beliefs of all patients as part of holistic health care. Hence, ovine-based collagen represents a new generation of collagen dressing that is widely sanctioned by all religious communities and cultures [10].

The collagen scaffold is proven to serve as a beneficial template to support cell growth and promote cell differentiation for tissue repair and regeneration [11,12]. These can be further supported with the ability of ovine-based collagen extracellular matrix (CECM) dressings to withhold the intricate collagen architecture of the indigenous tissue extracellular matrix (ECM) together with the ECM-associated secondary molecules comprised of laminin, fibronectin, and glycosaminoglycans [13]. The other advantage of ovine-based CECM dressing is that it can shield itself from noxious matrix metallopeptidase (MMP) activity present in the chronic wound microenvironment cushioned by its broad-spectrum buffering capacity for proteolytic collagenases and gelatinase [13]. Nevertheless, the downsides of collagen are mainly poor structural stability and mechanical properties [14,15]. An alternative of cross-linking the collagen may improve the tensile strength, and incorporation of additional biomaterial to form hybrid or composite ovine-based collagen dressing may further impart its mechanical characteristics.

Nowadays, biodegradable and environmentally friendly products have garnered worldwide attention. Compared to other natural biomaterials, nanocellulose has been used widely for tissue engineering due to its unique characteristics, such as large surface area and eco-friendliness [16]. Nanocellulose-based materials have been shown to have great biodegradability, biocompatibility, low-cytotoxicity, porosity, and desirable mechanical properties [17]. Various composites of nanocellulose have been proven to create synergistic effects for wound healing. A hybrid of nanocellulose and collagen exhibits low densities, high porosities, strong water absorption, and good mechanical strength [18,19]. By employing both collagen and nanocellulose, it was hoped that a better outcome can be achieved on wound dressing by extracting the benefits from both properties.

Among various nanocellulose sources, recent scientific research targeting the applications of wound dressings focuses on the use of bacterial cellulose, typically from the genera of *Gluconacebacter* and *Agrobacterium*. However, certain drawbacks are associated with bacterial cellulose, including high production and operating cost, use of expensive culture media, poor yields, and downstream processing [20,21,22]. This allows the plant-based nanocellulose to have an upper hand. Malaysia is a country well-endowed with palm oil. Indeed, Malaysia is the world’s largest producer of palm oil. 

In line with this, large biomass production occurs promises only 10% extraction rate with 90% biomass as byproduct [23]. Traditionally, these agricultural wastes were burnt, and the ash was recycled as a fertiliser, which has a negative impact on the environment. In addition, the lignocellulosic biomass waste from palm oil must be fully exploited to preserve the environment and sustainability of the palm oil industry. Hence, all these aspects trigger the curiosity to explore the potential of oil palm lignocellulosic biomass as an alternative renewable bioresource considering its great abundance and economic significance with respect to Malaysia specifically and globally. Therefore, this study aims to evaluate the physicochemical characterisation of a bilayer composite natural biomaterial as a potential wound dressing using ovine collagen and plant nanocellulose. A second aim is to examine the effect of these cross-linking agents for biodegradation and stability extension.

## 2. Materials and Methods

### 2.1. Collagen Extraction and Purification

Ethical approval was obtained from the Research and Ethical Committee of the Faculty of Medicine, Universiti Kebangsaan Malaysia (JEP-2019-389) prior to the start of the study. Collagen extraction and purification processes were conducted according to the previous study performed by Fauzi et al. (2016) with some modification [24]. Briefly, the source of the collagen in this study was extracted from ovine’s tendon. The fur, skin, fascia, and muscle tissue were thoroughly removed from the tendon before placing it for 48 h of the freeze-dried process. The tendon was cut and immersed in 0.35 M acetic acid (Radnor, PA, USA) at 4 °C for 24–48 h to dissolve the collagen for further extraction. The clear-coloured collagen was collected and amalgamated in sodium chloride (0.05 g/mL; Sigma-Aldrich, St. Louis, MO, USA). The solution is then incubated at 4 °C for 24 h prior to centrifugation at 10,000 rpm for 10 min. The collagen pellet was separated from the supernatant and dialysed for 72 h using a dialysis tube (molecular weight cut off = 14 kDa) (Sigma-Aldrich, St. Louis, MO, USA). with distilled water as the dialysis buffer. The distilled water was replaced every 12 h. Dialyzed collagen was again freeze-dried from 24 to 48 h and re-dissolved in 0.35 M acetic acid.

### 2.2. Nanocellulose Preparation

Oil Palm Empty Fruit Bunch Fiber (OPEFB) supplied by Szetech Engineering Sdn. Bhd. (Kampung Padang Jawa, Shah Alam, Malaysia) was used as the source of cellulose. The preparation of nanocellulose was done by the Faculty of Science and Technology, Universiti Kebangsaan Malaysia based on previous publications [25,26]. In brief, the OPEFB fibres were first subjected to soxhlet extraction using a 2:1 (*v/v*) ethanol/toluene solution (Sigma-Aldrich, St. Louis, MO, USA). for 6 h. It was then alkali treated with 4% (*w/v*) of sodium hydroxide (NaOH; Sigma-Aldrich, St. Louis, MO, USA).) solution. The white cellulose was obtained through a bleaching process that removes lignin and hemicellulose by sodium chlorite (NaClO_2_; Sigma-Aldrich, St. Louis, MO, USA) and glacial acetic acid (Sigma-Aldrich, St. Louis, MO, USA). The isolation of nanocellulose was carried out by acid hydrolysis with 60% (*v/v*) of aqueous sulphuric acid (H_2_SO_4_; Sigma-Aldrich, St. Louis, MO, USA) for 40 min at 45 °C. The resulting suspension was centrifuged for 10 min at 10 °C, dialysed against distilled water until a constant pH was reached, and ultrasonicated to disentangle the nanocrystal. Finally, the final product was freeze-dried to obtain a white powder and stored at 4 °C until further use.

### 2.3. Fabrication of Bilayer Hybrid Scaffold

Bilayer hybrid scaffold (BHS) was fabricated with the combination of two different distinct layers of nanocellulose and collagen. Briefly, the collagen solution with a concentration of 14.25 mg/mL was first pipetted into the desired mould and pre-frozen at −80 °C for 3 h as optimised by Fauzi et al. (2019) to form the bottom layer of the BHS [10]. For the control group, the top layer will be made of 1 mL collagen, hence the control BHS was solely made up of pure ovine collagen (POC). In contrast, the other treatment groups contain a mixture of collagen solution with different weights of OPEFB nanocellulose powder as the top layer. The different mixtures were 1 mL collagen with 1 mg nanocelulose (ColNc 1), 1 mL collagen with 5 mg nanocellulose (ColNc 5), and 1 mL collagen with 10 mg nanocellulose (ColNc 10) that were pipetted into a mould followed by the freeze-dried process for 24–48 h. The fabricated BHS was subsequently cross-linked with 0.1% genipin (Wako, Japan).

### 2.4. Water Absorption Capacity

The initial dry weight of the samples was determined (W_0_). The samples were then immersed in 2 mL of phosphate buffer solution (PBS) for 1 h at a room temperature of 37 °C. The excess water was removed by using a filter paper and the weight (W_1_) of the sample was recorded. The water absorption capacity was calculated as follow:(1)$$W_{s}\;\left(\%\right)=\frac{W_{1}-W_{0}}{W_{0}}\;\times \;100$$
where W_s_ is the water absorption capacity (%), W_1_ is the swollen weight (g), and W_0_ is the dry weight of the scaffold (g).

### 2.5. Porosity

The porosity of the scaffold was evaluated by solvent replacement method stated elsewhere [10] with some modification. The scaffold was immersed in 3 mL of absolute ethanol overnight and weighed after excess ethanol was discarded. The porosity was calculated using the equation below:(2)$$\mathrm{Porosity}\;\left(\%\right)=\frac{M_{2}-M_{1}}{\rho V}\;\times \;100$$
where M_1_ is the dry weight of the scaffold (g), M_2_ is the mass of the scaffold (g) after immersion in absolute ethanol, ρ is the density of the absolute ethanol, and V is the volume of the scaffold.

### 2.6. 3D-Microstructure

BHS sponge was prepared and visualised under scanning electron microscopy (SEM) using LEO 1450VP (Zeiss, Dublin, CA, USA). The freeze-dried BHS sponge was then dried using a critical dryer for 30 min and coated with gold in a sputter coater for 60 s using a sputter coating device Polaron SC 7680 (Polaron, London, UK). The samples were then sent for SEM viewing.

### 2.7. Biodegradation

The biodegradation rate of the bilayer scaffold was measured by the weight loss of the sample after a certain period. The initial weight (dry) of the samples was determined (W_0_) and the samples were then immersed in 2 mL 0.0006% (*w/v*) of collagenase (Worthington, Lakewood, NJ, USA) followed by an incubation step at 37 °C. The 0.6% (*w/v*) collagenase was prepared by dissolving the 0.24 g collagenase powder and further diluted at 1000× with phosphate buffer saline (PBS) to produce 0.0006% collagenase working solution.

After a certain period, the collagenase was removed from the samples. The samples were then rinsed gently with distilled water before freeze-dried. The weight (W_t_) was then recorded. The biodegradation property was calculated based on the days taken for all the samples to be fully degraded. The percentage for the weight loss of the scaffold was calculated using the equation below:(3)$$\mathrm{Mass}\;\mathrm{reduction}\;\left(\%\right)=\;\frac{W_{0}-W_{t}}{W_{t}}\;\times 100$$
where W_0_ is the dry weight and W_t_ is the final weight in a dry form.

### 2.8. Chemical Characterisation of Bilayered Hybrid Bioscaffold

EDX, FTIR, and XRD were used for chemical characterisation of BHS. The BHS was scanned using INCA Energy 2000 microscope (Oxford Instruments, Abingdon, UK) for EDX to characterise the main elements containing carbon, nitrogen, and nitrogen compositions of the analysed volume using EDX software.

To identify the effects of molecular fingerprint of the collagen in contrast to ColNc 1, ColNc 5, and ColNc 10 to the POC and nanocellulose powder. FTIR spectra were obtained from 650 to 4000 cm^−1^ using a Perkin Elmer Spectrum 400 FTIR Spectrometer (PerkinElmer, Waltham, MA, USA) with a Spotlight 400 Imaging System, in a wavenumber ranging from 650 to 4000 cm^−1^. The graph containing different peaks was analysed with the existing spectral database design according to the spectroscopy model. The structural information of the BHS on an atomic scale was obtained through XRD. The test was conducted using Bruker D8 Advance (Bruker, Hamburg, Germany) with CuKα radiation (λ = 1.54 nm) and a setting of 0.04° step size at 25 ℃ for 40 min. The subsequent data were analysed with the existing spectral database design according to the spectroscopy model.

### 2.9. Mechanical Strength

The mechanical properties including tensile strength, elastic modulus, and strain at failure were carried out using Instron^®^ Universal Testing Machine model 5557 (Instron, Norwood, MA, USA) equipped with Bluehill 3 software with a 500 N load cell and a tensile force applied at an extension rate of 5 mm min^−1^. The scaffolds thickness in the range of 1 mm were cut in the form of dumbbell-shaped specimens with 4 mm width, 20 mm gauge length, and 1 mm thickness. At least 3 specimens were tested to calculate the average value. The test was performed in a room condition.

### 2.10. Statistical Analysis

The data were analysed using GraphPad Prism version 7.0 (GraphPad Software, Inc., San Diego, CA, USA). The data collected from various quantitative parameters were presented as the mean ± standard deviation (SD) of the sample size. One-way analysis of variance (ANOVA) was applied to compare the control with multiple groups. The result was expressed as mean and standard deviation. A significant difference was considered when the *p*-value is < 0.05.

## 3. Results

### 3.1. Physical Characterisations of Hybrid Bioscaffold

#### 3.1.1. Water Absorption Capacity

The water absorption capacity of the scaffolds in PBS is shown in Figure 1. The water absorption capacity is an essential characteristic of an ideal wound dressing to maintain a moist environment to promote wound healing. The swelling ratio of all scaffolds was higher than 1500%. The POC exhibited the highest water absorption capacity with 2037.97 ± 125.94% followed by sample ColNc 10 (1993.12 ± 97.10%). The POC demonstrated a significantly higher water absorption capacity than ColNc 1 (1671.36 ± 33.39%) and ColNc 5 (1808.71 ± 74.28%). The water absorption capacity increased with the addition of nanocellulose percentage in all treated groups.

#### 3.1.2. Porous Microstructure

The porosity of the samples is as shown in Figure 2a. All samples exhibited good porosity with a result of more than 70%. The POC exhibits the best porosity with 78.67% ± 2.65% followed by ColNc 10 with 76.56% ± 2.35%. The other treated group, ColNc 1 and ColNc 5, recorded a porosity value of 73.44% ± 1.67% and 75.56% ± 0.89%, respectively. The POC demonstrated a significantly higher porosity compared to ColNc 1 and ColNc 5. The porosity of the treated group increased with the addition of nanocellulose percentage.

The morphology of the samples was studied under the scanning electron microscopy and the result showed that the BHS match the criteria of an ideal scaffold. It has interconnected pores with an average pore size ranging from 90 μm–140 μm. The pore size was also uniformly and normally distributed throughout the designated samples. The POC has the smallest pore size with a value of 97.17 μm ± 21.96 μm while ColNc 1 has the biggest pore size with a value of 139.12 μm ± 59.60 μm followed by ColNc 5 (119.94 μm ± 34.92 μm) and ColNc 10 (101.47 μm ± 59.60 μm). The pore size decreased with an increase of nanocellulose percentage in the treated group.

#### 3.1.3. Biodegradation

Figure 3 shows the mass reduction of bilayer hybrid bio-scaffold, which was determined by measuring the weight loss of the scaffold over a period of time via an enzymatic approach. The biodegradation rate of the POC is faster than those containing nanocellulose composition. The POC was fully degraded at day 18, which was the fastest among all samples, while the treatment group containing nanocellulose degraded at an average of day 20. ColNc 10 degraded at day 22, which was the slowest while ColNc 1 and ColNc 5 were fully degraded at day 19 and 20, respectively.

### 3.2. Chemical Characterisations of Hybrid Bioscaffold

#### 3.2.1. Energy Dispersive X-ray Spectrometry (EDX)

The principal behind EDX was an exhibition of the bombarded X-ray spectrum by detecting the existing elements from the POC, BHS and nanocellulose to obtain a localized chemical analysis. The rough areas will be identified by SEM. However, on the smooth surface, it is identified that the weight (%) from POC exhibited 44.34% carbon element compared to 19.31% of nitrogen and 36.34% of oxygen. The atomic (%) for carbon element was 50.28%, 18.78% for nitrogen and 30.94% for oxygen. Carbon showed to have the highest percentage of an atom compared to other elements due to its abundance. This is applied to all different nanocellulose samples concentrations as carbon showed the highest counts and energy from the graph compared to any other existing elements. The elements detected in nanocellulose were carbon, oxygen, and sulphur. As compared to POC and BHS, the carbon (53.3%) and oxygen (46.4) were in large weight percentage while the percentage of sulphur (0.4%) was small. Table 1 shows the elements found and its proportion in BHS and nanocellulose respectively.

#### 3.2.2. Fourier Transform InfraRed (FTIR)

The FTIR spectra were obtained from each sample of collagen, BHS, and nanocellulose as shown in Figure 4 The IR spectrum of POC showed absorbance of amide A due to NH stretching (3302 cm^−1^), amide B due to CH_2_ asymmetrical stretching (2926 cm^−1^), amide I due to NH bending (1631 cm^−1^), amide II due to CN stretching (1547 cm^−1^), and amide III (1237 cm^−1^). The displayed spectrum corresponds to the collagen characteristic, particularly collagen type 1. Meanwhile, the spectra captured from nanocellulose were O–H stretching at 3330 cm^−1^, CH_2_ bending at 1425 cm^−1^, C–O stretching at 1314 cm^−1^, while pyranose ring C–O–C skeletal vibrations were located at 1056 cm^−1^. These were the typical peaks found in nanocellulose. Whereas spectra observed in ColNc 1, ColNc 5, and ColNc 10 were quite similar with slight variations in between. Nevertheless, the composite BHS displayed all the common bands of which is the characteristic of collagen type 1.

Therefore, the FTIR spectra obtained showed there were no changes in the molecular structure of collagen and nanocellulose, respectively after a mixture of collagen and nanocellulose at different concentrations. Thus, it appeared that BHS has been successfully fabricated while maintaining the original characteristics from the functional groups.

#### 3.2.3. X-ray Diffraction (XRD)

XRD is one of the x-ray techniques available apart from X-ray absorption and X-ray fluorescence. The aim of using the XRD technique in this method is to determine the crystalline phases present in a material and thereby reveal chemical composition information. The molecular patterns obtained for BHS, pure collagen and nanocellulose powder are shown in Figure 5. The X-ray diffractograms show that the major intensity peak is located at 2θ value of around 22.5°, which is related to the crystalline structure of nanocellulose for ColNc 1 sample (Figure 5b). The highest intensity can be seen for ColNc which confirms the highly crystalline structure of ColNc compared to POC. Figure 5c–e clearly demonstrates that the crystallinity of material progressively increases with the addition of nanocellulose in the bilayer scaffold. The increase of crystallinity is expected to increase the mechanical strength and modulus of the bilayer scaffold.

#### 3.2.4. Tensile Strength

All scaffolds displayed a similar stress-strain pattern whereby the stress increased in a linear elastic relationship with respect to the applied strain. The Young’s modulus value was calculated from the stress–strain ratio observed at maximum load to determine the stiffness of the scaffolds in reflection of its mechanical strength. The POC demonstrated lower Young’s modulus 0.74 MPa ± 0.17 MPa as compared to the other bilayer composite scaffolds: 0.98 MPa ± 0.30 MPa in ColNc 1, 1.37 MPa ± 0.29 MPa in ColNc 5 and 1.49 MPa ± 0.21 MPa in ColNc 10. Thereby, this outlined that the incorporation of nanocellulose into the collagen can efficiently improve the mechanical properties of the bilayer composite scaffold. Among it, there was a significant improvement observed in ColNc 5 and ColNc 10 to the POC. The mechanical properties of the scaffolds, as in their elastic modulus, tensile strength, and strain at failure, are summarized in Table 2 and Figure 6.

## 4. Discussion

Wound dressing plays an important role in protecting the wounds from external contaminants, shielding the wound against mechanical trauma, enhancing the healing process as well as preventing complications such as gangrenous tissue, which may lead to amputation. Therefore, the proper construction of scaffold in tissue engineering is crucial in developing a good wound dressing for future use. A bilayer wound dressing is gaining much attention during this era of modernisation as both layers are made of different properties resembling closely to human skin. Each layer provides significant advantages as a wound dressing. In this study, a BHS was successfully fabricated using collagen from ovine and plant nanocellulose, which is a low-cost, abundant, and renewable material compared to cellulose from marine animals [27]. In order to affect the final structural, mechanical, and biological properties of collagen-based scaffolds, the BHS was all post-cross-linked with 0.1% (*w/v*) genipin to stabilise the scaffolds. Genipin is a natural cross-linker that has been proved to have low toxicity, improve mechanical strength, and demonstrate excellent biocompatibility as well as enhancing the fibroblast attachment [28,29,30]. When crosslinked to a collagen-based scaffold, it improves the mechanical strength and is resistance against enzymatic degradation [28,30]. Therefore, it enhances the property of the scaffold as a wound dressing. A better understanding of the physicochemical characteristics of scaffolds is pivotal in designing it according to the application for which it is required. An assessment of functional, thermal, and mechanical properties of scaffolds show clear indications of the limitations in which, the scaffolds can be developed without modifying the natural properties. Modifying the natural properties will greatly affect the physical or chemical processing.

The water absorption capacity results revealed that the scaffold exhibits excellent properties as a wound dressing. The crucial prerequisite to supply the nutrients and eliminate the metabolites for colonised cells is to ensure the scaffold has strong water absorption and retention capacity. Water absorption capacity is an important characteristic of an ideal wound dressing to maintain a moist environment. Moist wounds are known to heal faster due to the adequate supply of growth factors and other molecules to the healing tissues [31]. Moreover, it helps to keep excessive exudates [32]. A good swelling ratio is necessary to provide essential nutrients for the cells [33] and to deliver antibiotics to the wound bed [8]. There are few factors contributing to the swelling ratio property such as physicochemical properties of the biomaterials, the medium used to dissolve the scaffolds, surface wettability, and the use of cross-linker [34]. The application of cellulose in the sponge has increased the swelling ratio as nanocellulose exhibits higher water absorption capacity property and this is useful in wound care. The supplication of nutrients, which is needed by the colonised cells in the sponge will be entirely from the body fluid secreted by the wound. Therefore, in the process of constructing skin tissue engineering scaffold, it is necessary to evaluate water absorption and water uptake capacity [35]. Besides nanocellulose, collagen is well known to have good absorption capability in keeping the wound hydrated [36]. In this study, the same physicochemical properties were used for each scaffold as well as the same concentration of cross-linker, 0.1% genipin and PBS as the medium for each sample. Thus, the wettability of the scaffold plays an important role in water absorption capacity property. The POC showed higher hydrophilicity compared to other BHS with the highest rate of water absorption capacity of 2000%. The sponge-like scaffold shown by all samples is optimal for cell culture.

A scaffold with a highly porous structure can provide a large surface area that would allow adhesion and cell growth, cell distribution, and neovascularisation [37]. Open porous structures with interconnectivity are essential for nutrition, proliferation, and the formation of new tissue [38]. In general, interconnected porous scaffold networks allow the transport of nutrients, removal of wastes, and facilitate proliferation and migration of cells [39]. High porosity also enables the release of biofactors for nutrient exchange. Moreover, collagen promotes cellular motility, especially intense infiltration of neutrophils and inflammatory cells. These mediators actively invade the porous scaffold. A highly vascularised granulation tissue stimulates new granulation tissue and epithelial layers, thus promoting rapid skin recovery. Collagen-based implants are well known as vehicles to deliver keratinocytes and medication for skin replacement and burn-wound treatment. This is due to the ability of these implants to be infiltrated by amorphous connective tissue forming an extracellular matrix that is suitable for skin regeneration [8]. Nanocellulose has been proven to increase the porosity of the scaffold. A study reported that a higher concentration of cellulose gives a better porosity characteristic. The process of freeze-drying, which is also known as lyophilisation would be the simplest method used for the formation of interconnected pores in the scaffold [30]. Freeze-dry technique that was used in constructing these scaffolds allows the BHS to cool to the freezing point, which then separated in ice phase resulting in a porous structure [30]. The porosity reflects on the empty area that was occupied by the ice crystals. A porous scaffold exhibited by the samples can absorb large amounts of exudate, protect against microbial infection or growth, and maintain a clean and moist environment for wound healing [8]. The samples in this study have > 70% porosity value of which is the range of ideal wound dressing requirement within 60–90% porosity [40]. This would provide a large surface area to allow cell growth, cell distribution, and neovascularisation [37].

The result of good absorption capacity and high porosity was confirmed by SEM images that showed the scaffolds formed a three-dimensional (3D) porous structure. This BHS has good potential for wound dressing as it possesses a suitable porosity, pores size, and have interconnected pores that are very important for tissue repair. Interconnected pores exhibit by the scaffold mimic the architecture of our native skin layer, which helps in promoting the attachment, proliferation, and differentiation of fibroblast [41] along with efficient nutrient and oxygen supply to the seeded cells [42]. Pore sizes of 90–130 μm are sufficient to permit fibroblast migration and proliferation [43]. Another study stated that the average pore diameter from 20 μm to 125 μm was shown to be an optimal size for skin regeneration [44]. In this study, the pore size is in the range from 90 μm to 140 μm, which is consistent with other studies. The increments of nanocellulose concentration in the scaffold sample affect the pore size. The higher the concentration of nanocellulose in the scaffold, the smaller the pore size as observed using SEM. A smaller pore size scaffold has the ability to retain and hold more water after achieving its swelling equilibrium, thus maintaining persistent moisture needed for optimal wound healing. This proves that nanocellulose does help in defining the pore size in scaffold fabrication.

Biodegradation process plays a crucial role in wound dressing. Frequent change of dressing is not cost-effective and costs more burdens to the patients and/or caretakers. It is also important for remodelling of the connective tissues, regeneration, and mechanical support. Rapid degradation of a scaffold may affect the mechanical property while slow degradability may induce inflammatory reaction and will result in the failing of implantation on the skin [44]. Therefore, it is crucial to study the biodegradation property of skin tissue engineering. The ideal scaffold would be able to decrease inflammation, release medication, and help in the regeneration of tissue without needing to change frequently. The biodegradation data show that all samples possessed a good result. Samples were fully degraded by the enzyme collagenase after more than two weeks. This is optimal as frequent changing of the dressing can be avoided. The samples can gradually degrade and be replaced by new tissue from the adhered cells [45]. The presence of nanocellulose slows down and prolonged the degradation rate due to the absence of cellulase thus, suggesting that physical interaction between collagen and nanocellulose has greater hindrance effect. However, it will be dissolved slowly after all of the collagen has fully degraded since there was no more collagen to hold the cellulose. The scaffold has lost its structural properties and therefore, dissolved in the medium.

The values of elemental analysis in EDX look very similar even in the composite materials compared to the pure ovine collagen. Such a trend appeared to be inconsistent even with the increased of nanocellulose concentration. The possible limitation in this study was the lack of homogeneity of the nanocellulose in the solution due to the small amount has been introduced in this mixture. FTIR and EDX analysis of BHS was comparable with that of pure ovine collagen and pure oil-palm nanocellulose. FTIR analysis with mid-infrared wave number ranging 1630 cm^−1^–1230 cm^−1^ showed the presence of typical peaks of amide I, II, and III, which is the functional group for collagen [46], whereas in nanocellulose, it is represented by a pyranose ring with the wave band measured at 1056 cm^−1^. In BHS, it contains all the wave bands from both nanocellulose and collagen with a slight shift in terms of the location. This result was supported by the XRD analysis, which attributed to a diffused peak at about 2θ = 22.5° indicating the decreased intensity of the same peak. Moreover, previous study has demonstrated that the presence of collagen in the nanocellulose decreased the crystallinity index by disrupting the regular arrangement of nanocellulose molecular chains [47]. Therefore, it correlates with the diffraction peaks of the BHS that are similar to the pure collagen and nanocellulose, except to its property with a decrease in intensity. Together, all these results ascertain that the original functional features of both biomaterials are intact even after developing it into a composite scaffold.

Pure collagen scaffolds have always known to be associated with poor mechanical properties. Purcel et al. (2016) reported that the natural assembly structure was destructed during the extraction process regardless of the method [15]. Hence, intermolecular crosslinking was sought to modify the collagen scaffold’s mechanical strength and to preserve its stability, thus correcting such limitation [15]. Studies showed that collagen scaffold crosslinked with genipin have a better mechanical profile without having its porosity properties affected and have minimal toxicity [48]. This synergistic improvement suggests good chemical compatibility between the two phases, as well as good dispersion [49]. Since the mechanical properties of the scaffold are often mandated by the microstructure of its constituent materials, another improvement that can be considered would be by blending the collagen with other material such as nanocellulose as in this study [30,50,51]. The mechanism was illustrated by Zhijiang and Guang (2011) whereby when polymer pairs exist in two phases, the mechanical properties of a composite material may be dominated by the polymer-rich phase [47]. In this study, the collagen–nanocellulose would be the polymer-rich phase to form a continuous matrix, and the collagen layer being the secondary phase to reinforce the matrix by stress transfer between the interfaces. The result of the spectacular increase of elastic modulus and tensile strength in the bilayer hybrid composite could be explained by the easier penetration of nanocellulose in the collagen matrix and impregnation of both individual fibrils and bundles.

## 5. Conclusions

The bilayer hybrid scaffolds were successfully fabricated using the freeze-drying technique and showed satisfying physicochemical characterisations. Along with its large surface area from the well-interconnected pore network structure, the scaffolds have good swelling properties and demonstrate adequate resistance against degradation in vitro. The decreased crystallinity index of the bilayer hybrid scaffold makes it more flexible and improve its mechanical property. Taken together, these results highlighted the potential utilisation of oil palm nanocellulose and ovine-based collagen into a bilayer hybrid scaffold for biomedical application in wound dressing. Nevertheless, further research is required to study cell adhesion, proliferation, as well as biocompatibility to ensure that the scaffold is suitable and safe for human use.

## Figures and Tables

**Figure 1 materials-13-04352-f001:**
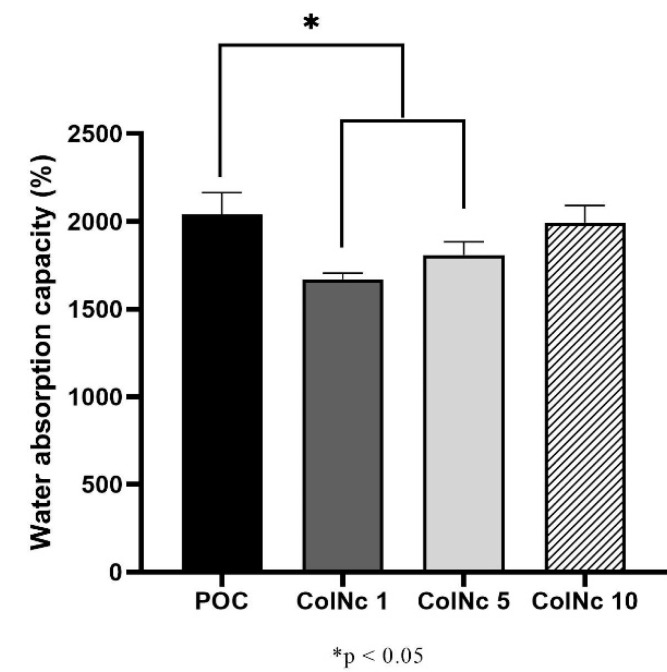
The water absorption capacity (%) of bilayer hybrid scaffold fabricated by different concentration of nanocellulose. The water absorption capacity increased with increasing concentration of nanocellulose in the treated group while POC remained as the highest among all of the samples. (*) represents a significant difference (*p* < 0.05; *n* = 3) between control group, POC.

**Figure 2 materials-13-04352-f002:**
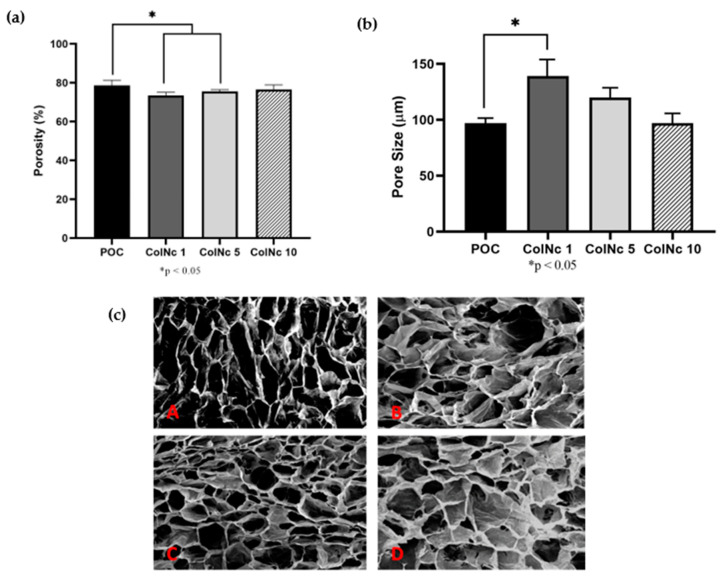
Effects of different concentration of nanocellulose on (**a**) The porosity of bilayer hybrid scaffold. (**b**) The pore size. (**c**) The cross-sectional SEM images of the scaffold at 50X magnification where (A) POC (Control); (B) 1 mL collagen with 1 mg nanocellulose (ColNc 1); (C) 1 mL collagen with 5 mg nanocellulose (ColNc 5); (D) 1 mL collagen with 10 mg nanocellulose (ColNc 10). (*) represents a significant difference (*p* < 0.05; *n* = 3) between control group, POC. The porosity of scaffolds increased while the pore size decreased with higher concentration of nanocellulose.

**Figure 3 materials-13-04352-f003:**
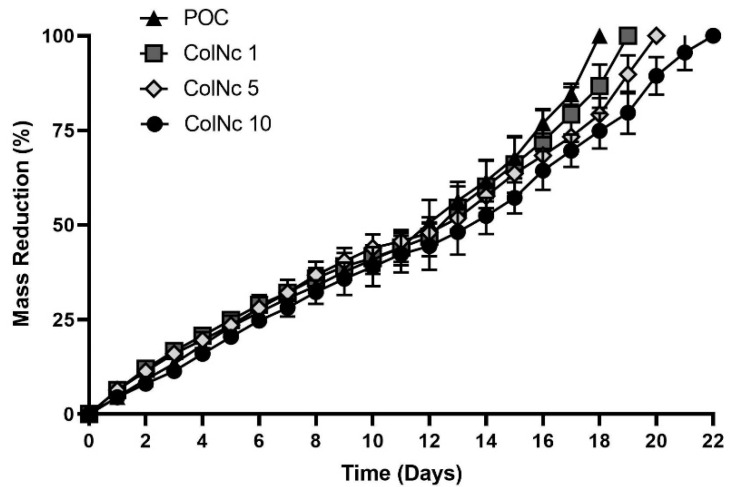
The biodegradation rate of bilayer hybrid scaffold was compared between POC (Control), 1 mL collagen with 1 mg nanocellulose (ColNc 1), 1 mL collagen with 5 mg nanocellulose (ColNc 5) and 1 mL collagen with 10 mg nanocellulose (ColNc 10). The average biodegradation rate for treated group was 20 days while POC was fully degraded at day 18.

**Figure 4 materials-13-04352-f004:**
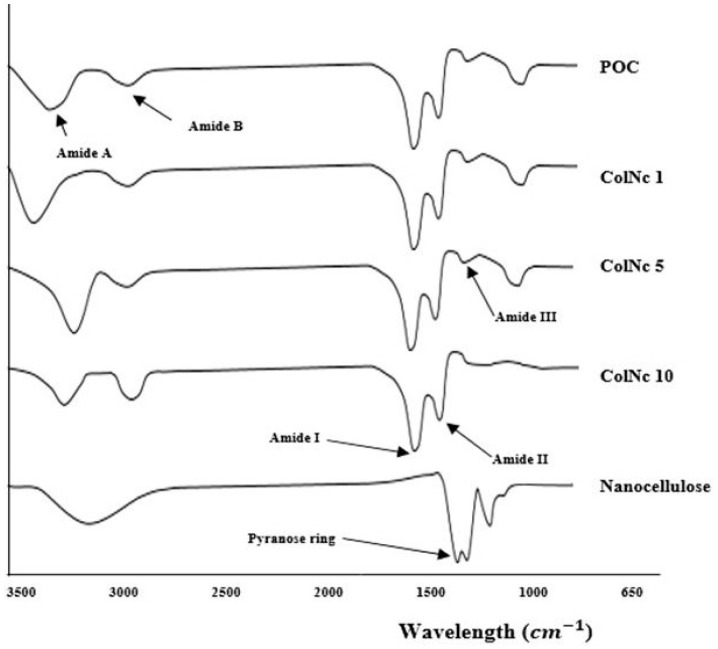
The chemical structure of fabricated bioscaffolds via FTIR. The various absorbance peaks represented by different chemical structure including amide A, amide B, amide I, II and III.

**Figure 5 materials-13-04352-f005:**
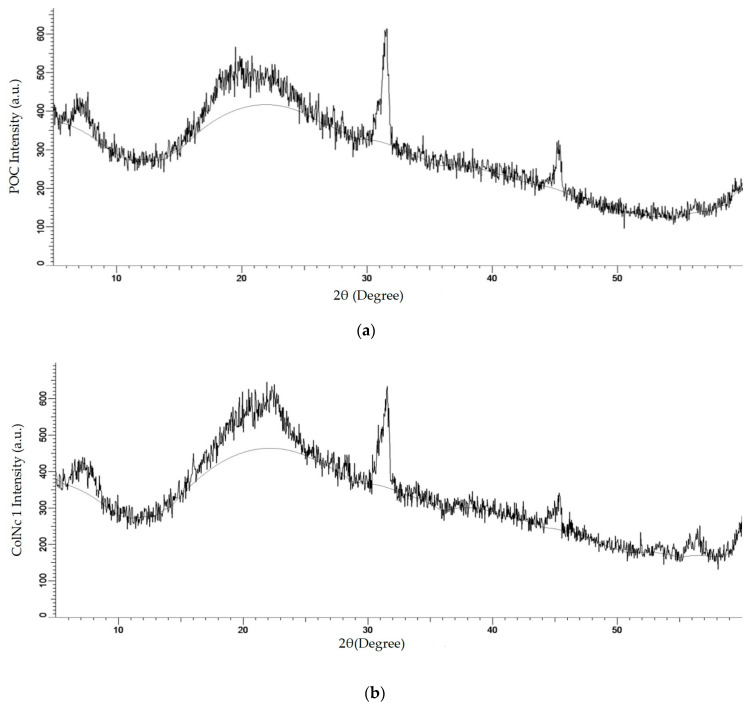
The XRD molecular pattern of (**a**) POC (Control); (**b**) 1 mL collagen with 1 mg nanocellulose (ColNc 1); (**c**) 1 mL collagen with 5 mg nanocellulose (ColNc 5); (**d**) 1 mL collagen with 10 mg nanocellulose (ColNc 10); (**e**) nanocellulose.

**Figure 6 materials-13-04352-f006:**
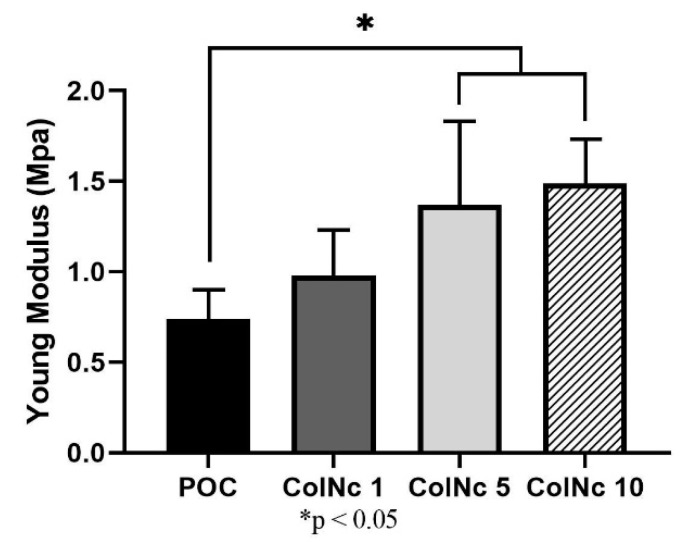
Mechanical profile of pure ovine collagen (POC) against various concentration of composite BHS. Study proved that mechanical strength varies proportionally with concentration of nanocellulose in the BHS. (*) represents a significant difference (*p* < 0.05; *n* = 3) with POC group.

**Table 1 materials-13-04352-t001:** Elemental analysis by EDX where (**a**) shows percentage by weight and (**b**) shows percentage by atomic. Elements detected in both POC and BHS scaffolds were carbon, oxygen, and nitrogen with carbon being the most abundant element. Whereas, elements detected in nanocellulose were carbon, oxygen and Sulphur.

**(a)**					
**Type of BHS** **Element**	**POC**	**ColNc 1**	**ColNc 5**	**ColNc 10**	**Nanocellulose**
**Weight (%)**					
Carbon	44.34	47.77	45.60	46.26	53.3
Nitrogen	19.31	15.79	18.55	13.45	–
Oxygen	36.34	36.44	35.85	40.29	46.4
Sulphur	–	–	–	–	0.4
**(b)**					
**Type of BHS** **Element**	**POC**	**ColNc 1**	**ColNc 5**	**ColNc 10**	**Nanocellulose**
**Atomic (%)**					
Carbon	50.28	53.88	51.57	52.55	60.37
Nitrogen	18.78	15.27	17.99	13.10	–
Oxygen	30.94	30.86	30.44	34.36	39.46
Sulphur	–	–	–	–	0.17

**Table 2 materials-13-04352-t002:** Mechanical strength of bilayer hybrid scaffold was evaluated through Young’s Modulus and Tensile Strain. Result has showed increment in tensile strength and tensile strain with increasing concentration of nanocellulose. This study has showed that ColNc 10 has demonstrated the best result out of all scaffold.

Sample	Modulus [MPa]	Tensile Strength at Break [MPa]	Tensile Strain (Extension) at Break [%]
POC	0.74 ± 0.17	0.37 ± 0.07	67.15 ± 2.48
ColNc 1	0.98 ± 0.30	0.28 ± 0.04	39.04 ± 8.17
ColNc 5	1.37 ± 0.29	0.42 ± 0.11	47.04 ± 0.95
ColNc 10	1.49 ± 0.21	0.57 ± 0.04	50.72 ± 10.89

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
