# Peer review of "Physicochemical Characterization of Bilayer Hybrid Nanocellulose-Collagen as a Potential Wound Dressing"

_materials, 2020, doi:10.3390/ma13194352_

Round 1
Reviewer 1 Report
The authors present a manuscript on the physicochemical characterisation of bilayer hybrid nanocellulose-collagen as potential use for wound dressing material.
Abstract poorly written with almost every sentence grammatically poor.
Introduction starts off well, but references needed for points in line 54, line 96, line 102, etc.
Some sentences need minor grammatical corrections (e.g., deviates from normal…, chronic wounds bring…, following guidance from Fauzi et al. (2016), sodium chloride…, pure collagen scaffolds have…).
Line 196 – not clear what is meant by ‘scaffolds in PBS was stated’…
Results 3.1.3 – it is unclear from the Methods how long the degradations are going on for – here in the Results it mentions at least 22 days, but I do not see information on this in the Methods.
Figure captions should better describe what they are, e.g., Fig. 4 is presumably EDX spectrum, and Fig. 5 FTIR, but this is only inferred through the section subheadings when it should be clear in the captions.
Reviewer 2 Report
Shen et al. present a prospective investigation of their synthesis and characterization of novel bilayer hybrid nanocellulose-collagen platforms that could serve as wound dressings. For a variety of religious, social and economic/environmental reasons the authors chose sheep tendon and palm tree products as the sources of collagen and nanocellulose, respectively.
The authors provide a very detailed, step-by-step description of the processing and manufacture of pure collagen (control) and three different combinations of collagen and nanocellulose products.
Why a concentration of 14.25 mg/ml of ovine collagen?
The mixtures were chemically crosslinked by genipin in premade molded shapes prior to a gauntlet of physical assessments.
The tests included the ability to absorb water, assessments of tensile strength, electron microscopic assessments of porosity, etc. Chemical composition was confirmed via energy-dispersive x-ray spectroscopy (EDX), Fourier transform infrared spectrometry (FTIR), and X-ray diffraction (XRD). The ability to be degraded by dilute collagenase over time was also assessed.
The results are presented in a very clear fashion, and they demonstrate that the various composites are as good or superior to the control condition. The 1:5 and 1:10 combinations of collagen:nanocellulose appear viable as candidates for future testing as biodegradable dressings.
Aside from frequent grammatical errors, I have little else to criticize about this interesting paper with promising results.
Reviewer 3 Report
PHYSICOCHEMICAL CHARACTERIZATION OF BILAYER HYBRID NANOCELLULOSE-COLLAGEN AS A POTENTIAL WOUND DRESSING written by Kai Shen Ooi et al dealt with preparation and characterization of a composite of Nanocellulose and collagen, having potential in wound dressing. The paper could potentially be considered appropriate to the Material journal, but actually it cannot be published in the present form. Some parts seems to be missed and some other very poorly described. Herewith attached some major concerns regarding the manuscript: - The use of nanocellulose obtained from residue of palm oil manufacture is totally missed. How did you get nanocellulose from a so different source? Please implement with these informations. - 2.2. Fabrication of bilayer hybrid scaffold needs some amendments: "A mixture of collagen with palm tree-based nanocellulose (obtained from Faculty of Science and Technology, Universiti Kebangsaan Malaysia) in the ratio of 1:1 (ColNc 1), 1:5 (ColNc 5) and 1:10 (ColNc 10) was then pipetted into each mould respectively followed by the freeze-drying for 24-48 hours." Please specify the nature of the ratio (volume, weight, or what else?). Be more precise: 1:1, 1:5 and 1:10 is referred to Collagen:cellulose? If not, please correct. - Please more detail concerning collagenase (L159). -Cocerning the presentation of experimental data Control was referred to Pure ovine Collagen sponge experimentally isolated. It would be more appropriate to stress this info into all figures captions. In addition, in order to better compare the effect of addition of nanocellulose also the parameters related to pure nanocellulose should be added: water adsorption capacity, porosity, 3D microstructure, biodegradation, mechanical strength, elemental analysis. They could give more options to discuss about the effect of the composite materials. - Values of elemental analysis actually look to be very similar even in the commposite materials: as they were made by collagen only. Please comment. - Figures 4, 5 and 6 needs to be improved: all these figures need to be replaced by new ones having an adequate quality. - FTIR look to be as cellulose was totally absent in composite system. Please comment. - in the abstract it was reported that: "The fabricated biomatrix was post-cross-linked with 0.1% (w/v) genipin 29 (GNP)." Such a post-cross-linking did not appear in the paper, unless a final citation as a work conducted by others... Please revise this aspect: implement with experimental data or eliminate this sentence into the abstract. L83 collagen extracellular matrix (CECM) should be more used: the authors after use ECM acronym; L110-111: ....the potential of oil palm as an alternative renewable bioresource.... please correct.Author Response
Please see the attachment

Round 2
Reviewer 3 Report
I read the revised version of manuscript "PHYSICOCHEMICAL CHARACTERIZATION OF BILAYER HYBRID NANOCELLULOSE-COLLAGEN AS A POTENTIAL WOUND DRESSING" by Kai Shen Ooi et al.
The manuscript resulted improved, but unfortunately some further discrepancies were evidenced: in L150-155 were reported that 1 ml of collagen (14.25 mg of collagen, L147) were dispersed with 1, 5 and 10 mg of nanocellulose, whereas in L249-250, L262-262, it was reported that the ratio "collagen vs nanocellulose" was 1mg vs 1, 5 and 10 mg of nanocellulose in ColNc1, ColNc5 and ColNc10 respectively.
Please correct.
In any case, sample ColNc10 at least should give different analysis: FTIR, elemental analysis and XRD should be different.
Whereas they look too similar. In the first round of revision I asked for elemental analysis of pure nanocellulose, and possibly to introduce also the respective physical determinations.
Please implement also these points.
Round 3
Reviewer 3 Report
All requests of amendments were considered in this version.
Even XRD graphs resulted significantly improved.
Please, the last suggestion is referred to FTIR spectra: it would be preferable to report spectrum as reported in the "Author response" file. Fig. 5 in the manuscript looks to be badly manipulated, on the countrary "Fig.2 The re-run FTIR for ColNc 10" reported in the "Author response" file looks much real and more conceivable for a publication in "Materials".
Author Response
Please see the attachment.
Thank you for the suggestion. We had replaced the FTIR results at Line 312-315.